# COVID-Vaccines in Pregnancy: Maternal and Neonatal Response over the First 9 Months after Delivery

**DOI:** 10.3390/biom14040435

**Published:** 2024-04-03

**Authors:** Alice Proto, Stefano Agliardi, Arianna Pani, Silvia Renica, Gianluca Gazzaniga, Riccardo Giossi, Michele Senatore, Federica Di Ruscio, Daniela Campisi, Chiara Vismara, Valentina Panetta, Francesco Scaglione, Stefano Martinelli

**Affiliations:** 1Neonatal Intensive Care Unit, ASST Grande Ospedale Metropolitano Niguarda, 20161 Milan, Italy; alice.proto@ospedaleniguarda.it (A.P.); stefano.martinelli@ospedaleniguarda.it (S.M.); 2Department of Medical Biotechnology and Translational Medicine, Postgraduate School of Clinical Pharmacology and Toxicology, University of Milan, 20122 Milan, Italy; stefano.agliardi@unimi.it; 3Department of Oncology and Hemato-Oncology, University of Milan, 20122 Milan, Italy; arianna.pani@unimi.it (A.P.); francesco.scaglione@unimi.it (F.S.); 4Department of Biomedical, Surgical and Dental Sciences, Postgraduate School of Microbiology and Virology, University of Milan, 20122 Milan, Italy; silvia.renica@unimi.it (S.R.); federica.diruscio@ospedaleniguarda.it (F,D,R.); 5Chemical-Clinical and Microbiological Analyses Unit, ASST Grande Ospedale Metropolitano Niguarda, 20161 Milan, Italy; riccardo.giossi@ospedaleniguarda.it (R.G.); michele.senatore@ospedaleniguarda.it (M.S.); daniela.campisi@ospedaleniguarda.it (D.C.); chiara.vismara@ospedaleniguarda.it (C.V.); 6L’altrastatisticasrl, Consultancy & Training, Biostatistics Office, 00174 Rome, Italy; valentina.panetta@laltrastatistica.com

**Keywords:** breastfeeding, COVID-vaccines, neonatology, pregnancy, pharmacology, SARS-CoV-2

## Abstract

Vaccination against SARS-CoV-2 has been demonstrated to be safe during gestation. Nevertheless, there are no robust data investigating the entity of maternal antibodies’ transmission through the placenta to the newborn and the persistence of the antibodies in babies’ serum. The objective of this study is to assess the maternal antibody transmission and kinetics among newborns in the first months of life. Women having received one or two doses of anti-SARS-CoV-2 mRNA-vaccines during pregnancy at any gestational age, and their newborns, were recruited and followed-up over 9 months. Ninety-eight women and 103 babies were included. At birth, we observed a significant positive correlation between maternal and neonatal serum anti-SARS-CoV-2 antibody levels and a significant negative correlation between the time since last dose and antibody levels in mothers with two doses. Over the follow-up, the birth antibody level significantly decreased in time according to the received doses number at 3, 6, and 9 months. During the follow-up, we registered 34 dyad SARS-CoV-2 infection cases. The decreasing trend was slower in the SARS-CoV-2 infection group and among breastfed non-infected babies. Antibodies from maternal anti-SARS-CoV-2 vaccination are efficiently transferred via the placenta and potentially even through breast milk. Among newborns, antibodies show relevant durability in the first months of life.

## 1. Introduction

Severe Acute Respiratory Syndrome Coronavirus 2 (SARS-CoV-2) is a highly contagious respiratory virus that quickly emerged as a global public health emergency, impacting millions of individuals [1,2,3,4,5]. Over the past few years, several vaccines have been developed to combat the SARS-CoV-2 pandemic, with a particular focus on protecting vulnerable populations, including pregnant women. It is well-documented that pregnant women who contract the virus are at a higher risk of experiencing severe symptoms and mortality compared to non-pregnant adults. This is due to the physiological and immunological changes that occur during pregnancy, which make them more susceptible to complications [6,7,8]. The risk of neonatal complications, such as premature delivery, meconium staining, respiratory distress, and perinatal death, also increases [8,9]. Although pregnant women were excluded from the pivotal trials of COVID-vaccines, a large subsequent observational study reported comforting data on efficacy and safety of vaccinating against SARS-CoV-2 in this setting [10]. However, especially at the beginning of vaccination campaigns, the lack of data on vaccination during pregnancy contributed to vaccine hesitancy and increased the risk of SARS-CoV-2 infection exposure. Currently, the literature indicates that pregnant women exhibit strong immune responses to COVID-19 mRNA vaccines, achieving antibody titers comparable to those of non-pregnant women of reproductive age [11,12,13,14], with similar safety and reactogenicity profiles [12,13,14,15].

To date, the impact of maternal anti-COVID vaccination on newborns is still partially unclear. Our study aims to assess antibody transfer in babies born to women vaccinated against SARS-CoV-2 during pregnancy, the antibody titer correlation in the dyad, and the antibodies’ durability. Second, we also aimed to evaluate the role of breastfeeding, with the hypothesis that breastfed children maintained higher antibody titers over time. Moreover, the impact of post-natal SARS-CoV-2 infection on antibody titer trends was evaluated among babies. Safety and tolerability information were also collected during the medical checks.

## 2. Materials and Methods

### 2.1. Participants Recruitment

This is a prospective observational study involving a cohort of pregnant women and their respective infants. Recruitment took place at ASST Grande Ospedale Metropolitano Niguarda (Milan, Italy), with a follow-up conducted over 9 months. All included women received at least one dose of mRNA vaccine against SARS-CoV-2 at any trimester of pregnancy. Trimesters were defined as follows: First (1st–12th week of gestational age), Second (13th–25th week of gestational age), and Third (26th week of gestational age-delivery). Eligibility criteria also included age > 18 years, willingness to participate, and provision of informed consent. Exclusion criteria comprised maternal age < 18 years, lack of informed consent, previous SARS-CoV-2 infection, or patients undergoing blood transfusions. Upon hospital admission for delivery, all women underwent routine confirmation of negative SARS-CoV-2 status through nasopharyngeal swab reverse transcription-polymerase chain reaction and screening for specific serologic evidence (anti-N positivity) of past infection before enrollment, along with the collection of anamnestic information. Additionally, a study questionnaire was administered to assess information regarding pregnancy, timing of COVID-19 vaccine doses, and the type of vaccine received. Demographic and clinical data were collected upon recruitment.

The study was conducted in accordance with the Declaration of Helsinki (as revised in 2013). This study also received approval from the ASST Grande Ospedale Metropolitano Niguarda Milano Ethics Commission Board and the national ethics committees for COVID-19 studies at Istituto Nazionale per le Malattie Infettive Lazzaro Spallanzani IRCCS Roma.

### 2.2. Study Design and Sample Collection

Maternal and neonatal peripheral blood samples were collected at birth (T0), at 3 months (T1), 6 months (T2), and 9 months (T3) after delivery (Figure 1). The first blood sampling occurred prior to the initiation of breastfeeding, immediately following delivery.

Follow-ups for all participants with a previously detectable antibody titer were scheduled at 3 months +/− 15 days, 6 months +/− 15 days, and 9 months +/− 15 days. Samples were centrifuged to obtain sera, stored at +4 °C until processed, aliquoted into cryogenic vials, and stored at −80 °C. Serum samples were analyzed with the SARS-CoV-2 IgG II Quant assay, quantitative, and the SARS-CoV-2 IgG assay, qualitative, (Alinity SARS-CoV-2 IgG assay, Abbott Diagnostics, Chicago, IL, USA) [16]. These tests use Chemiluminescent Microparticle Immuno-Assay (CMIA) technology that detects IgG antibodies directed against the receptor-binding domain (RBD) of the Spike protein and IgG antibodies directed against the nucleocapsid protein of SARS-CoV-2, respectively. Anti-RBD antibodies can be detected in both infected and vaccinated patients, while anti-N antibodies are detectable only in the sera of previously infected individuals. Blood samples from all mothers and babies involved in these cases of infections were tested to evaluate the serological increase in anti-S antibodies titer and to detect the presence of anti-N antibodies in the dyad. SARS-CoV-2 infection was defined as a positive PCR for SARS-CoV-2, a positive SARS-CoV-2 antigen rapid test, or the appearance of newly detected anti-N serum antibodies at the follow-up visit. Children were not routinely tested for SARS-CoV-2 (antigen rapid test or PCR), but they were still included in the COVID-group in the case of a mother’s infection, as close contacts. Samples were processed and results were considered positive with an index (S/C) > 1.4 for SARS-CoV-2 IgG and a value > 50.0 AU/mL for SARS-CoV-2 IgG II Quant.

### 2.3. Statistical Analyses

Categorical variables were expressed as absolute numbers and percentages, while continuous variables were summarized by mean and standard deviation (sd) or median and first and third quartile (Q1–Q3). Correlations between the levels of antibodies in mothers and children and between the levels of antibodies (in both mothers and children) and temporal variables were calculated with Pearson’s regression coefficient. The Kruskal–Wallis test was used to compare antibodies between the three groups formed by considering the trimester of vaccination. Post hoc analysis was conducted using the Mann–Whitney test adjusted with the Bonferroni correction. To evaluate children’s antibodies over time, the mixed-model Tobit regression was used. Log values were used as the dependent variable, and time and groups (if present), and their interactions, were the fixed factor. The random factors were the mother and child code. The geometric mean and its standard error were reported for each time point. Percentage change from birth and its CI95% were estimated. Stata 16.2 software was used for the analysis, and a *p*-value < 0.05 was considered statistically significant.

## 3. Results

We approached 100 pregnant women who were expected to meet inclusion criteria based on the patient characteristics provided and collected upon delivery room admission. Two of them were subsequently excluded due to anamnestic incompatibilities: One woman underwent SARS-CoV-2 infection during pregnancy, while one had not received any dose, contrary to what was initially communicated. A total of 98 women were therefore included in the study who had been admitted for delivery to the Niguarda Hospital of Milan, and whose demographics and clinical characteristics are presented in Table 1.

All the included women received vaccination against SARS-CoV-2 with mRNA-vaccines. In particular, 82 (83.7%) women received BNT162b2 (Comirnaty) vaccine, and 16 (16.3%) received COVID-19 Moderna mRNA-1273 (Spikevax) vaccine. Moreover, 24 (24.5%) out of 98 received just one dose, while the other 75 (75.5%) completed the two-doses vaccination cycle. Considering the last vaccination trimester, 11 out of 98 women (11.2%) received a vaccination in the first trimester of pregnancy, 7 (7.1%) in the second, and 80 (82.7%) in the third one.

Among mothers who were followed up, 56 (57.1%) women received a booster dose after delivery. In particular, 25 out of 56 women received the Comirnaty vaccine, and the other 31 received the Spikevax vaccine. According to the primary cycle, 28 women underwent a homologous booster vaccination, while 28 underwent a heterologous vaccination. A mean increase of +625.86% (sd 118.53) in antibody titers was recorded among women who received the booster dose (Appendix A).

Moreover, 103 babies were included (93 single pregnancies and 5 twin pregnancies). According to information collected from mothers on each follow-up visit, 78 of these (83%) were breastfed (any breastfeeding), and in most cases, breastfeeding was maintained for at least 3 to 6 months.

Over the 9-month follow-up period, 75 (76.5%) mothers and 79 (76.7%) babies were checked at T1, 57 (58.2%) mothers and 61 (59.2%) babies at T2, and 42 (42.9%) mothers and 46 (44.6%) babies were able to attend the last follow-up visit at T3. In nine cases, the dyad was not admitted to the subsequent follow-up step because the baby serum was negative for SARS-CoV-2 antibodies at the previous check; one newborn was excluded, as per exclusion criteria, because she had received a blood transfusion. Other withdrawals occurred for personal reasons.

During the follow-up, we registered 34 cases of SARS-CoV-2 infection among mothers; there were no clinically significant differences in the baseline characteristics of infected compared to non-infected groups (Appendix A). In 12 (35%) cases, the infection occurred before the booster dose. According to indications, the booster dose was withheld in 10 of these 12 cases due to this circumstance; however, in the remaining two instances, the booster dose was administered based on clinical considerations. Additionally, in the other 22 (65%) cases, the infection occurred after the booster dose had been administered.

No serious adverse events were reported by mothers following vaccination. Furthermore, the whole cohort of our babies presented good outcomes after delivery and over the follow-up period (Table 1).

Moreover, no cases of infection resulted in severe symptoms (i.e., dyspnea, fever > 39 °C) or hospitalizations, either among mothers or, especially, among children.

Anti-RBD antibodies were found in both mothers (99%) and newborns (98%) at T0. In particular, only one mother and two newborns in the one-dose group tested negative for serum anti-RBD antibodies. We found a significant positive correlation between maternal serum levels of SARS-CoV-2 antibodies and neonatal serum levels at birth (*p* < 0.001) (Figure 2A) and a significant negative correlation between time elapsed since last-dose administration and maternal serum IgG levels in mothers (*p* = 0.032). This correlation was present in mothers with two doses (*p* < 0.001) (Figure 2B), but not in mothers with one dose (*p* = 0.615).

In Figure 2A, a positive and significant correlation between the antibodies of mother and child at birth can also be observed, both in mothers with two doses and in mothers with one dose (overall *p* < 0.001; one-dose group *p* < 0.001; two-doses group *p* < 0.001). There was no significant correlation between time elapsed since the last-dose administration and children’s serum IgG levels (overall *p* = 0.169; one dose *p* = 0.834; two doses *p* = 0.100) (Figure 2C).

In Figure 3A,C, the relationships between mother and child antibodies and gestational age (GA) at vaccination are represented.

Mothers vaccinated during the first trimester of pregnancy and their newborns showed the lowest serum antibody levels (Appendix A). The post hoc comparison confirmed that mothers who received a vaccination in the third trimester (Figure 3B) had higher antibodies than women vaccinated in the other two trimesters (*p* < 0.001). These differences were significant between the first and second trimester (*p* = 0.008) and between the first and third quarters (*p* = 0.001). The differences in infant antibodies between trimesters (Figure 3D) were also significant (*p* = 0.004).

Data for children over time were available for 79 babies (4 twin couples) at T1, 61 babies at T2, and 45 babies at T3. It can be seen that the antibodies level at birth decreases significantly over time (*p* < 0.001) (Figure 4).

The birth antibody level decreased with a significant difference over time (*p* < 0.001) according to the received doses number at 3 (*p* < 0.001), 6 (*p* = 0.013), and 9 (*p* < 0.001) months (for estimated mean value and percentage changes from birth, see Appendix A). If we compare antibody kinetics in the two groups of infants (infection vs. no-infection) (Figure 4A), we can observe that, in the infection group, the decline in antibody titers was significantly slower (*p* < 0.001) (Appendix A).

Regarding the role of breastfeeding, if we consider lactating women only in the no-infection group (Figure 4B), we observed a slower decline in antibody titers in totally breastfed babies compared to partially breastfed and formula-fed babies (see also Appendix A).

## 4. Discussion

Recent research has indicated the possible transfer of anti-SARS-CoV-2 antibodies across the placenta following maternal mRNA COVID-19 vaccination. This suggests that maternal vaccination might offer a degree of protection to newborns at birth [5,9,17,18,19,20,21,22]. However, the duration for which these protective antibodies remain present in the baby’s serum remains unclear [5]. Although the primary goal of antenatal SARS-CoV-2 vaccination is to prevent maternal illness, it is as yet uncertain what the most effective immunization regimen might be (including vaccination timing and the number of doses) to sustain maternal immunity throughout gestation and safeguard the offspring [23]. Rottenstreich et al. confirmed the efficient transplacental transfer of SARS-CoV-2 antibodies after vaccination, with persistent anti-RBD specific IG detected at 3 months in all the studied infants and higher antibody concentrations following third-trimester vaccination [23]. Moreover, preliminary studies demonstrated significantly higher maternal and neonatal SARS-CoV-2 IgG antibodies levels at birth after a second-trimester booster (third dose) of maternal Comirnaty (BNT162b2) COVID-19 vaccination compared with the primary two-dose vaccination series [22,24], with a higher efficacy against SARS-CoV-2 variants, including the B.1.617.2 (delta) and the B.1.1.529 (omicron) variants, via the hybrid immunity phenomenon [25].

According to Shook et al., about 94% of newborns born to vaccinated mothers exhibit detectable serum SARS-CoV-2 spike protein IgG at 2 months of age, decreasing to around 60% by 6 months. Surprisingly, only 8% of infants born to mothers infected with SARS-CoV-2 but not vaccinated during pregnancy showed detectable antibodies. This suggests robust vaccine-induced protection with antibody longevity that appears to surpass that shown by infants born to mothers previously infected with SARS-CoV-2 [26]. In addition, Nir et al. demonstrated notably higher antibody levels in both maternal and cord blood samples from vaccinated women compared to non-vaccinated individuals who experienced COVID-19 during pregnancy [20,27].

Furthermore, more recent studies confirmed vaccine-induced protection from COVID-19 in early infancy, with SARS-CoV-2 antibodies levels enhanced by breastfeeding until at least 6 months of age in infants whose mothers were vaccinated during pregnancy [28] and the effectiveness of maternal vaccination during pregnancy against COVID-19 hospitalization in infants aged < 6 months [29].

Our study provides a uniquely prolonged clinical and serological follow-up (9 months) in a relatively wide cohort of patients, not only stating the detection of antibody levels in newborns born to mothers exposed to anti-SARS-CoV-2 vaccination but also investigating the potential impact of important post-delivery variables, such as early-life SARS-CoV-2 infection and breastfeeding, on antibody kinetics. In particular, our study demonstrated the efficient transfer of SARS-CoV-2 IgG across the placenta from women vaccinated during pregnancy, with a positive correlation between maternal and neonatal serum antibody concentrations. Hence, in addition to maternal protection against COVID-19, vaccines may provide neonatal immunity, while humoral response is still inefficient. Vaccination against SARS-CoV-2 during pregnancy offers important neonatal protection by transplacental antibody passage, leading to a significant risk reduction for this category, correlated with strict adherence to the current epidemiological measures.

While the study indeed demonstrates a robust antibody response in mothers and the transfer of antibodies to newborns, it is crucial to acknowledge that it primarily explores the standard vaccination regimen typically used in adults in that phase of the COVID pandemic. Therefore, while the findings are promising, they do not fully explore alternative vaccination schedules that may potentially offer enhanced protection. Hence, further investigation into alternative vaccination strategies is required to evaluate the most efficacious approach for safeguarding both maternal and neonatal health. This could involve assessing varying dosing intervals, alternative vaccine formulations, or novel adjuvants to optimize the immunization regimen.

Even if all the participants presented detectable serum antibodies, mothers vaccinated during the first trimester of pregnancy and their newborns had the lowest serum antibody levels, indicating that newborns might better benefit from maternal antibodies if vaccination occurs later in pregnancy (second or third trimester). The variation in antibody levels between mothers vaccinated in different trimesters can be influenced by the duration of antibody production, the timing of vaccination relative to delivery, and the half-life of antibodies. It has been confirmed that the transplacental transfer of antibodies progressively increases starting from the second trimester of pregnancy, a period of time in which the placenta becomes increasingly permeable. In this way, the antibody half-life allows for the protection of the newborn in the first months of life, a period in which the newborn is immunologically more vulnerable. However, the exact mechanisms and extent of antibody transfer during pregnancy and the persistence of antibodies in newborns are areas of ongoing research in the field of maternal immunology and perinatal health [30]. Moreover, these findings can be understood in the context of immunomodulatory changes that occur in the second trimester of pregnancy that favor maternal tolerance of the developing fetal semi-allograft and promote an immunological quiescence state [11]. Additionally, maternal cytokines play a crucial role in regulating the immune response during pregnancy and may influence fetal development and programming. Following COVID-19 mRNA vaccination during pregnancy, maternal cytokines produced in response to the vaccine could potentially cross the placenta and affect the fetal immune system. On the other hand, the presence of passively transferred cytokines/antibodies influences the cytokine secretion ability of splenocytes in the neonate, which provides novel evidence that maternal immunization can influence the newborn’s cytokine milieu and may impact immune cell differentiation (e.g., Th1/Th2 phenotype). Therefore, these maternally derived cytokines may play an essential role, both as mediators of early defense against infections and possibly as modulators of the immune repertoire of the offspring, but also in determining newborn’s susceptibility to allergic and autoimmune diseases later in life [31]. While the exact impact of maternal cytokines from COVID-19 mRNA vaccination on fetal development is not fully understood, ongoing research aims to investigate the potential implications for infant health and immunity.

Data regarding vaccines for influenza, tetanus toxoid, reduced diphtheria toxoid, and cellular pertussis-Tdap suggest that the placental transport system selectively transfers IgG antibodies. Antibody transfer is minimal before 16 weeks but increases throughout the second trimester, peaking in the third trimester to provide even higher neonatal antibody titers. This process is not fully understood but may be due to cytotropic expression and the neonatal-Fc receptor [5]. While there are similarities in the transfer of antibodies from vaccinated individuals to newborns across different vaccines, there are also notable differences in the specific antigens targeted, the mechanisms of antibody transfer, and the duration of passive immunity provided to infants; some vaccines may confer longer-lasting immunity to newborns due to the persistence of transferred antibodies, while others may require additional booster doses for sustained protection [32]. The antibodies produced in response to tetanus toxoid vaccination, as well as those produced after the reduced diphtheria toxoid vaccine, do not cross the placenta efficiently. Therefore, maternal antibodies transferred to the fetus are minimal, providing limited protection to newborns. Furthermore, while maternal antibodies generated by pertussis vaccinations do transfer across the placenta, they decline rapidly in the infant, providing only short-term protection. Newborns are susceptible to pertussis until they receive their primary vaccination series. To protect vulnerable infants, vaccination of pregnant women with Tdap during each pregnancy is recommended, preferably between 27 and 36 weeks of gestation. Additionally, the level and duration of antibody transfer may vary depending on factors such as the timing of vaccination during pregnancy and the type of vaccine administered; therefore, further research is needed to better understand the dynamics of antibody transfer for each vaccine and optimize vaccination strategies to protect maternal and child health [32,33,34].

Having a higher antibody level at birth could relate to a lower risk of severe complications in cases of infections among newborns. Amid our infants exposed to SARS-CoV-2, only mild symptoms and no hospitalizations were reported. Moreover, comparing the antibody kinetics, we observed that, in the infection group, the decline in antibody levels was significantly slower. This can be realistically related to the SARS-CoV-2 infection that occurred during the follow-up period, showing a certain capability, even by the immune systems of few-months-old babies, to generate antibodies in response to a SARS-CoV-2 infection. This does not mean that a protective role of maternal antibodies can in any case be crucial in guaranteeing a certain level of protection, particularly in the first and most fragile weeks of life. However, the differential immune response to SARS-CoV-2 variants in maternal immunity involves a complex interplay between antibody transfer, T-cell response, breastfeeding, and vaccination status. While maternal immunity can provide some degree of protection against SARS-CoV-2 variants, the effectiveness of this immunity may vary depending on factors such as the specific mutations present in the variants and the vaccination status of the mother [35].

Furthermore, it should be considered that, while COVID-19 mRNA vaccines are generally safe and effective, they can cause mild to moderate local and systemic reactions, as well as rare, adverse events such as myocarditis, pericarditis, and thrombosis with thrombocytopenia syndrome (TTS). It is important to note that the benefits of COVID-19 vaccination for pregnant individuals and their babies generally outweigh the risks, as vaccination reduces the risk of severe illness, hospitalization, and adverse outcomes associated with COVID-19 infection during pregnancy [36,37].

Antibody levels in children have a decreasing trend during the first months of life, but antibodies can be detected until 9 months after birth. For this reason, vaccination in pregnancy may have a protective role in offspring, as well as in pregnant women. Likewise, antibodies may be detected in breast milk [28,38,39], and it is possible that breastfed babies may show a slower drop in antibody serum titers.

## 5. Conclusions

Vaccination is currently the most important intervention to protect pregnant and breastfeeding individuals from COVID-19-related morbidity and mortality. The lack of pregnancy-specific safety and efficacy data during the initial vaccine rollout resulted in inconsistent guidance from multiple authorities including public health, regulatory, and professional societies, ultimately delaying vaccine access [10,40]. An additional maternal COVID-19 vaccination benefit may be the transfer of maternal immunity to newborns and to the youngest babies, for whom a vaccine is not available yet [5]. Likewise, the data reviewed herein suggest that SARS-CoV-2 binding antibodies from maternal vaccination are efficiently transferred via the placenta. Moreover, vaccinations in the second/third trimester induced greater immunogenicity in mothers when compared with first-trimester vaccinations.

Anti-SARS-CoV-2 antibodies retain a strong neutralizing capacity; therefore, anti-COVID vaccination during pregnancy should be strongly recommended as soon as possible, at any gestation time, in order to maximize pregnant women’s protection. Additionally, offering booster doses, particularly during the third trimester, can extend protection to the offspring [5,38]. In addition, a protective role of breastfeeding may be present [28,38,39].

However, several questions remain. What are the thresholds of protection in infant cord blood and breast milk? How does transferred immunity vary with the timing of vaccination relative to delivery and breastfeeding duration? What is the immunity durability in infants and how does it protect them from infection?

Further research is needed to reinforce public health policy regarding vaccination during pregnancy and determine several unanswered questions.

## Figures and Tables

**Figure 1 biomolecules-14-00435-f001:**
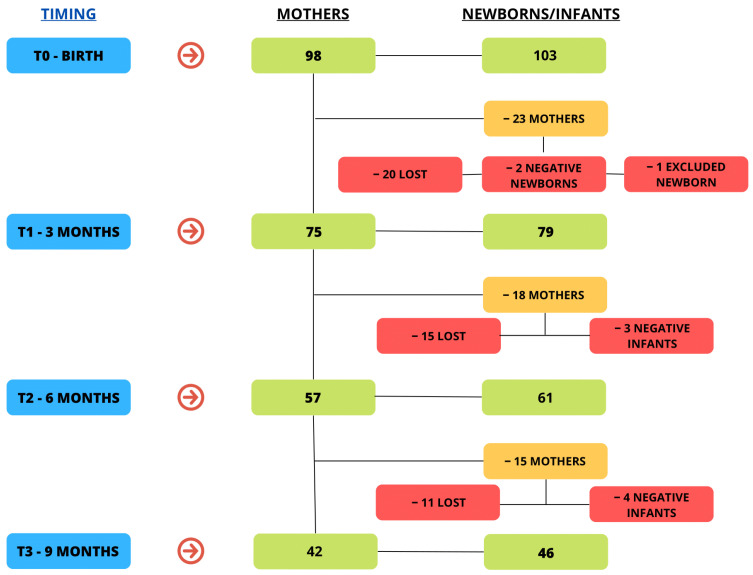
Study population. We included 98 pregnant women and their respective newborns (five twin pregnancies). Forty-six dyads were lost during the 9-month follow-up.

**Figure 2 biomolecules-14-00435-f002:**
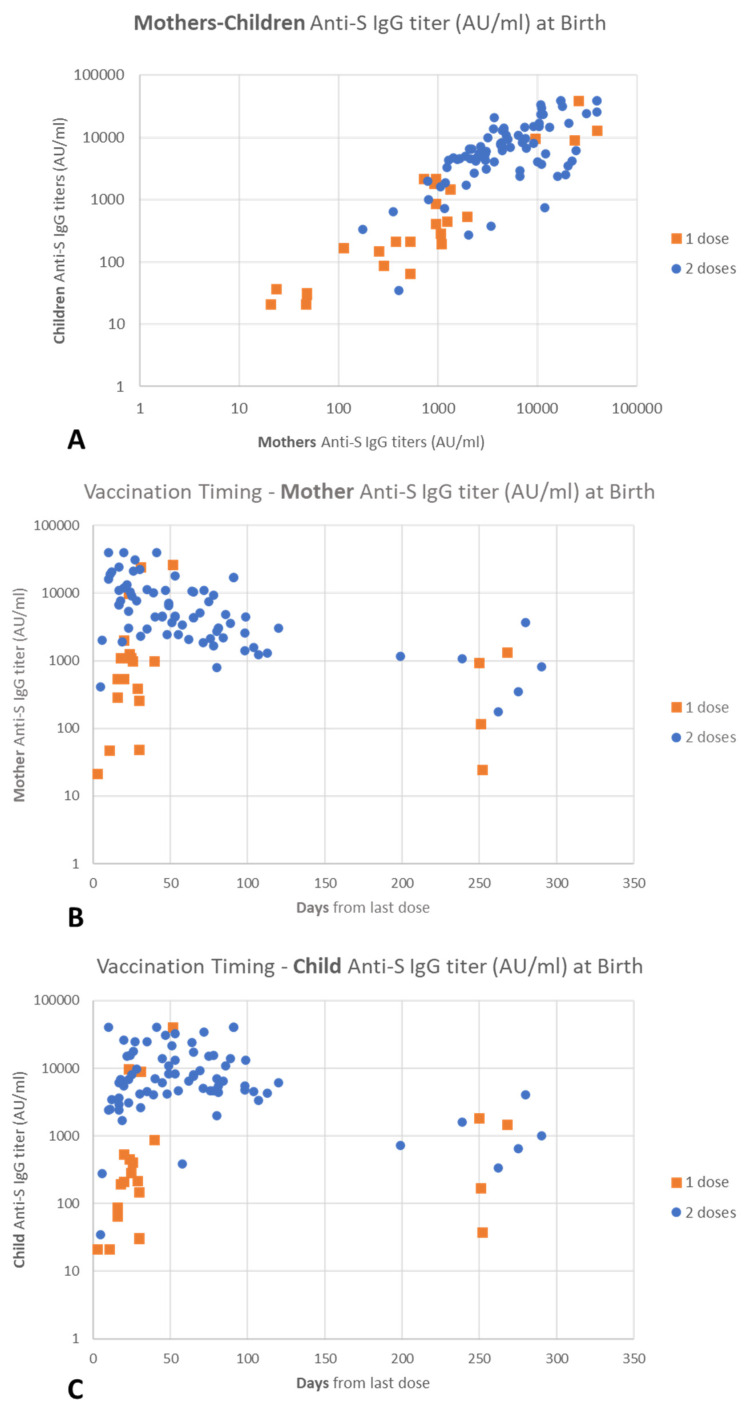
Antibodies at T0 (Birth Time). Orange: one-dose group (*n* = 24), Blue: two-doses group (*n* = 74): (**A**) correlation of maternal/neonatal anti-S IgG titer according to number of doses received by the mother; (**B**) association between maternal anti-SARS-CoV-2 specific serum IgG and distance (days) from last-dose administration; and (**C**) association between neonatal SARS-CoV-2 specific serum IgG and distance (days) from last-dose administration.

**Figure 3 biomolecules-14-00435-f003:**
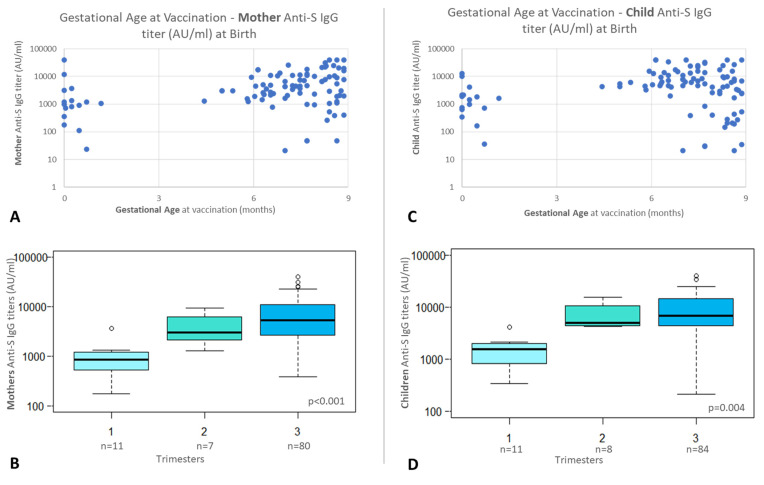
Antibody titers according to vaccination timing: (**A**) mothers’ antibody titers according to time of vaccination (*n* = 98); (**B**) mothers’ antibody titers according to gestational trimester of vaccination (*p* < 0.001); (**C**) newborns’ antibody titers according to time of vaccination (*n* = 103); and (**D**) newborns’ antibody titers according to mothers’ gestational trimester of vaccination (*p* = 0.004).

**Figure 4 biomolecules-14-00435-f004:**
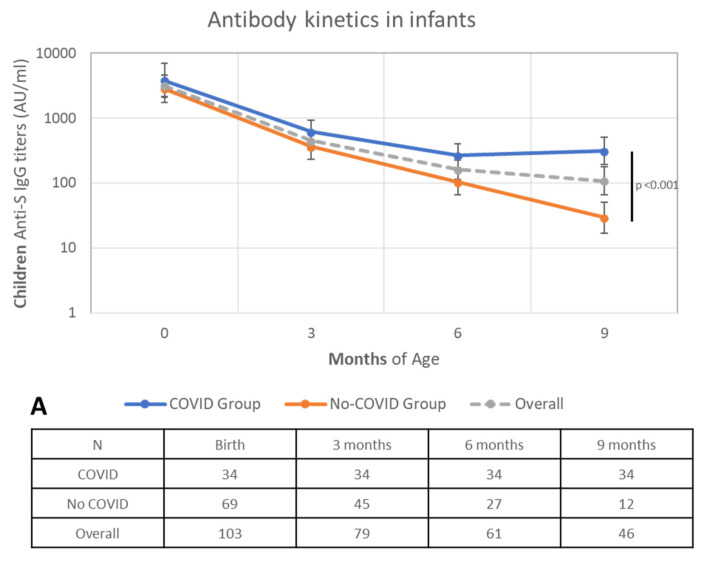
Antibody titers kinetics in infants (geometric mean values) over the 9-month follow-up period: (**A**) antibody kinetics in infants: COVID Group vs. No-COVID Group vs. Overall; and (**B**) antibody kinetics among No-COVID Group infants according to type of feeding: formula-fed vs. mixed-fed vs. breastfed infants. N values are reported in the Table below each graph.

**Table 1 biomolecules-14-00435-t001:** Mothers’ and babies’ demographics and clinical characteristics.

**Mothers**				
T0			*n* = 98	
Age	(mean, sd)		35.23	5.41
	(median, Q1–Q3)		35.50	32–39
Race	(*n*, %)	Caucasian	93	94.90
		Asian	1	1.02
		Hispanic	2	2.04
		North African	2	2.04
Number of doses	(*n*, %)	1	24	24.49
		2	74	75.51
GA 1st dose (days) ^1^	(mean, sd)		175.91	80.04
	(median, Q1–Q3)		196	153–238
GA 2nd dose (days) ^2^	(mean, sd)		203.93	66.80
	(median, Q1–Q3)		218	189–248
GA at childbirth	(mean, sd)		272.91	11.93
	(median, Q1–Q3)		275	268–280
2nd dose post-partum	(*n*, %)		23	23.47
Type of delivery	(*n*, %)	ED	73	74.49
		CS	21	21.43
		OVD	4	4.08
COVID	(*n*, %)		34	34.69
3rd dose	(*n*, %)	After childbirth	56	57.14
Comorbidity	(*n*, %)		13	13.27
Maternal diseases	(*n*, %)		13	13.27
Adverse Events to vaccination	(*n*, %)	No	81	82.02
		Yes	17	19.10
Admitted to follow-up	(%)		*n* = 75 (73.5)	
SARS-CoV-2 cases among followed-up	(*n*, %)		34	25.5
			*n* = 34	
		Period of Infection		
	(*n*, %)	0–3 months	10	29.41
		3–6 moths	15	44.12
		6–9 months	9	26.47
	(*n*, %)	Before 3rd dose	12	35.29
		After 3rd dose	22	64.71
**Babies**				
T0			*n* = 103	
Sex	(*n*, %)	Male	61	59.22
		Female	42	40.78
Weight (g)	(mean, sd)	Birth	3056.21	512.48
	(median, Q1–Q3)		3080	2780–3400
		3 months ^3^	5785.41	841.75
			5950	5000–6450
		6 months ^4^	7485.56	906.59
			7400	6800–8000
		9 months ^5^	8530.00	940.51
			8050	7900–9000
Lenght at birth (cm)	(mean, sd)		48.61	2.73
	(median, Q1–Q3)		49	48–50
Head circumference at birth (cm)	(mean, sd)		33.82	1.48
	(median, Q1–Q3)		34	33–35
Weight at birth (cent)	(mean, sd)		38.74	28.07
	(median, Q1–Q3)		33	14–60
Lenght at birth (cent)	(mean, sd)		36.95	27.18
	(median, Q1–Q3)		33	13–53
Head circumference at birth (cent)	(mean, sd)		43.06	27.92
	(median, Q1–Q3)		37	20–69
APGAR 1′	(*n*, %)	10	30	29.13
		9	58	56.31
		8	11	10.68
		≤7	4	3.88
APGAR 5′	(*n*, %)	10	85	82.52
		9	15	14.56
		8	3	2.91
Feeding (T0)	(*n*, %)	Formula-fed	15	16.13
		Mixed-fed	23	24.73
		Breastfed	55	59.14
Admitted to follow-up	(%)		*n* = 79 (81.5)	

GA = gestational age; ED: eutocic delivery; CS: Caesarean section; OVD: operative vaginal delivery. ^1^ *n* = 93; ^2^ *n* = 71; ^3^ *n* = 74; ^4^ *n* = 5; ^5^ *n* = 10.

## Data Availability

The data presented in this study are available on request from the corresponding author.

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
