# Peer review of "COVID-Vaccines in Pregnancy: Maternal and Neonatal Response over the First 9 Months after Delivery"

_biomolecules, 2024, doi:10.3390/biom14040435_

Round 1
Reviewer 1 Report
Comments and Suggestions for Authors
This study focuses on the transmission of maternal antibodies to the fetus during pregnancy that then persist in the newborn and later. This is studied in relation to the timing and number of vaccinations against SARS-CoV-2. Babies were studied up to 9 months of age. Transfer of antibodies was investigated as crossing the placenta and tested in newborns. The levels of antibodies over time up to 9 months of age in these babies was also investigated in relation to breastfeeding. The studies shows transmission of antibodies to the fetus during pregnancy and also some evidence of transmission through breast milk that are durable and appear to be protective.
This is an interesting study with a relatively large number of mother-child dyads. However, there are some revisions and clarifications that are needed including a specific statement of the novelty of this work in comparison to already published work in this area that the authors do provide in their discussion.
Major comments
1. Please explicitly state what is novel about these results compared to the studies listed in the discussion that seem to have already shown what this study is showing. The authors describe their results on Page 10 lines261 and 262 but this seems to just repeat what is already published. Novelty of this study should be very carefully described.
2. Please confirm that blood was taken from the newborns before first breast feed that could have transferred antibodies.
3. Of the 34 followed up who were diagnosed with COVID during pregnancy, were there any differences among the women or baby characteristics i.e. Table 1 compared to those who did not contract the infection? This would be important to know given changes in antibodies could be impacted by physiological differences in the mothers and/or babies. Should the table be separated into those who had COVID and those who did not, rather than just all grouped together.
4. Please provide more explanation for the 10 dyads for whom there was considerably more days elapsed since the last vaccine dose. Could these 10 dyads be ones that had COVID and so did not require a booster immunization?
5. Please provide n values for all graphs. Did Figure 2 include the women who contracted COVID during pregnancy? In Figure 3, what were the n values for each of these gestational ages? Also please define the definition of trimesters in the methods.
6. Not sure I understand the reasoning behind removing the 9 cases where the baby serum was not positive for antibodies at the previous check. Wouldn't this be important information to provide on the graph? – clearly there was a reduction of these antibodies – was this related to lack of breastfeeding in these women, i.e. is it likely that these antibodies transfer during breastfeeding so could the explanation for no antibodies be that the woman stopped breastfeeding and these antibodies cleared?
7. In the conclusion the authors indicated that “vaccination in the first trimester induced greater immunogeneicity in mothers when compared with second-trimester vaccination”. But the authors show in Figure 3 that the IgG titre is the lowest when mothers are vaccinated in the first trimester? So how does this relate to greater immunogenicity?
8. Which results indicate that breastfeeding may increase Ab transfer to the babies? It does not seem to have been included. If in supplementary, perhaps this should be part of the main part of the paper if this is a major component?
Minor comments
1. Abstract – the second sentence talks about transmission of antibodies to the newborn. However, it seems that the information provided in this study are not only looking at potential transmission to the newborn and child through breastfeeding but talking about transmission across the placenta during pregnancy that results in antibodies in the newborn. Please revise to be more accurate.
2. Introduction page 2, lines 52-54 – the first sentence indicates that efficacy and safety to vaccinate pregnant women against SARS-CoV-2 has been demonstrated; however, the next sentence states that pregnant women were excluded from COVID vaccine trials – these two statements seem to contradict each other. Please revise.
3. The second figure is misnumbered as Figure 1.
4. Page 7 line 190-192 – suggest moving this to where the figure was first discussed previously. It seems out of order here.
5. Significant differences should be reflected in the graphs, not only reported in the results section, e.g. Figures 3 and 4.
6. Supplementary table 1 – please define the before as before (pre) and after (post) to ensure that the results in the table are understandable.
7. Supplementary table 2 explanation does not make sense. These levels are not the levels of the mother and child in the three trimesters. These were measured at birth as I understand it and these are the values separated by when the mother was vaccinated – were these first vaccinations or any vaccination? More explanation to make this clear is needed here.
8. Supplementary Tables 3-5 – require n values.
9. Were any statistics performed on supplementary results? Nothing is indicated in the legends to these tables. If this was done, then this should be indicated within the tables. If this was done and there were no significant differences, then this should also be indicated.
Comments on the Quality of English LanguageA recommendation to review the use of English language in the manuscript. Some changes are needed.
Reviewer 2 Report
Comments and Suggestions for Authors
Overall, I appreciate the study design, having a good sample size for such an important study and longitudinal sampling at different time points are strengths of this study.
While the study by Proto et al provides valuable insights into the transfer of anti-SARS-CoV-2 antibodies from vaccinated mothers to newborns, there are several limitations and shortcomings that should be at least considered/discussed:
The study notes the uncertainty regarding the most effective immunization regimen, including vaccination timing and the number of doses, to sustain maternal immunity throughout gestation. Further research is needed to determine the optimal vaccination strategy for maximizing protection for both mothers and newborns.
While the study mentions higher efficacy against SARS-CoV-2 variants, including the delta and omicron variants, additional research is required to comprehensively assess the effectiveness of maternal vaccination against a broader range of variants that may emerge in the future. At least the authors should discuss the differential immune response to SARS-CoV-2 strain/variants (PMID: 37676005). Thus, such difference may influence maternal immunity.
The study suggests that vaccination during the second or third trimester may result in higher antibody levels in newborns, but the underlying immunomodulatory changes during pregnancy are complex and not fully understood. A more detailed exploration of these changes and their impact on antibody transfer is warranted. Do the authors consider that the variation in antibody levels between individuals vaccinated in the first trimester compared to the third trimester might be attributed to the temporal gap, leading to a reduced transfer of antibodies to the newborns? The antibody half-life?
The study briefly mentions data regarding vaccines for influenza, tetanus toxoid, reduced diphtheria toxoid, and cellular pertussis-Tdap but does not delve into a detailed comparison. A more comprehensive analysis of the similarities and differences in antibody transfer between various vaccines could strengthen the study.
The study mentions the potential protective role of breastfeeding but does not provide additional side effects of vaccinations against SARS-CoV-2.
The potential adverse effects of COVID-19 mRNA vaccines (PMID: 35537987, PMID: 37112659) and its impact through the transfer of cytokines should be at least mentioned/discussed. Maternal cytokines can be transferred to the newborn with both positive and negative effects (PMID: 28167667).
In relation to the above point, it would be highly informative to assess the presence of cytokines (multiplex) the same way done for antibodies. This is not a required but suggestion. If the authors are unable to quality cytokines/chemokines it should be at least mentioned as described above.
In conclusion, while the study contributes valuable information, these limitations highlight extensive details on how maternal immunization may influence antibody levels in infants.
Round 2
Reviewer 1 Report
Comments and Suggestions for Authors
This study focuses on the transmission of maternal antibodies to the fetus during pregnancy that then persist in the newborn and later. This is studied in relation to the timing and number of vaccinations against SARS-CoV-2. Babies were studied up to 9 months of age. Transfer of antibodies was investigated as crossing the placenta and tested in newborns. The levels of antibodies over time up to 9 months of age in these babies was also investigated in relation to breastfeeding. The studies shows transmission of antibodies to the fetus during pregnancy and also some evidence of transmission through breast milk that are durable and appear to be protective.
The authors have carefully considered and answered all questions/concerns. However, there are just some minor follow up comments and suggestions.
Minor suggestions
1. Thank you for including Table S2. Could you please comment in your text as to whether there were any significant differences between these two groups? It does not look like it but the readers need to know that the analysis was done and there were no differences observed. This then implies that the women who were infected after delivery in the follow up period along with their babies did not differ. Given that these 34 dyads are included in the results, I think doing these statistics and reporting any differences (or not) is important.
2. Tables in Figure 4 - please just define in the figure legend that these represent the n values for the graphs.
3. The authors indicate that no statistical analyses were done on the information provided in the supplementary tables. In some cases, statistical analyses could provide more information than the tables alone especially since there are statements within the text that refer to the results in the supplementary tables e.g. In Table S3, the authors indicate that the lowest levels of antibodies were found in women who received vaccinations in the first trimester. To state that the levels in first trimester are different from the other trimesters requires statistical analyses. Please do the analyses for Table S3 and carefully consider whether other information provided in the supplementary results should also be statistically analyzed to strengthen statements of the findings
Reviewer 2 Report
Comments and Suggestions for Authors
Congrats for the great work! The authors sufficiently addressed my concerns, which resulted in the improvement of their manuscript.
Author Response
Thank you for your appreciation and the useful suggestions provided.